# Fast and Accurate Factual Inconsistency Detection Over Long Documents

**Barrett Martin Lattimer**
ASAPP
blattimer@asapp.com

**Patrick Chen**
ASAPP
pchen@asapp.com

**Xinyuan Zhang**[*]
Abstractive Health
dylanz0426@gmail.com

**Yi Yang**
ASAPP
yyang@asapp.com

## Abstract

Generative AI models exhibit remarkable potential; however, hallucinations across various tasks present a significant challenge, particularly for longer inputs that current approaches struggle to address effectively. We introduce SCALE (**S**ource **C**hunking **A**pproach for **L**arge-scale inconsistency **E**valuation), a task-agnostic model for detecting factual inconsistencies using a novel chunking strategy. Specifically, SCALE is a Natural language inference (NLI) based model that uses large text chunks to condition over long texts. This approach achieves state-of-the-art performance in factual inconsistency detection for diverse tasks and long inputs. Additionally, we leverage the chunking mechanism and employ a novel algorithm to explain SCALE's decisions through relevant source sentence retrieval. Our evaluations reveal that SCALE outperforms existing methods on both standard benchmarks and a new long-form dialogue dataset ScreenEval we constructed. Moreover, SCALE surpasses competitive systems in efficiency and model explanation evaluations. We have released our code and data publicly to GitHub[1].

## 1 Introduction

Large Language Models (LLMs) have shown immense promise in various applications, but deploying them in real-time presents certain challenges such as hallucinations (Cao et al., 2018; Falke et al., 2019; Kryściński et al., 2019; Fabbri et al., 2021a; Honovich et al., 2022). Hallucinations, or factual inconsistencies generated by a model relative to a source document, can mislead the user and undermine trust in LLMs. Thus, detecting factual inconsistency in LLM generations is crucial for the future of LLMs, especially with the growing popularity of platforms like ChatGPT.

Prior research on inconsistency detection has predominantly dealt with short documents in offline settings (Laban et al., 2022; Schuster et al., 2022; Utama et al., 2022) and relied heavily on sentence-level text matching techniques. Consequently, these methods exhibit slow performance in processing longer documents and suffer from poor calibration. Such characteristics pose substantial challenges in implementing them in real-world online environments, where incorporating inconsistency detection could potentially result in a substantial increase in latency. Additionally, the absence of well-calibrated scores complicates the balancing act between mitigating the risk of incorporating hallucinations and excluding pertinent information from the model output. Given the exponential growth in context sizes (maximum allowed tokens of an input) of contemporary large language models (LLMs),[2] there is an increasing urgency to develop efficient and effective approaches for inconsistency detection in lengthy documents.

In addressing the challenges, we introduce SCALE (**S**ource **C**hunking **A**pproach for **L**arge-scale inconsistency **E**valuation), a method designed for efficient detection of factual inconsistencies in generated sentences by identifying related source text snippets. SCALE consists of two crucial components. First, it builds on a Natural language inference (NLI) based method, integrating a novel chunking mechanism for rapid and accurate online performance in diverse natural language generation (NLG) tasks. Second, model explanation is essential for real-time deployment of inconsistency detection systems, facilitating swift human inspection to determine model configurations. We show that our chunking mechanism improves calibration scores and enables the use of a binary search tree algorithm for rapidly locating relevant source text snippets for a target sentence, ultimately enhancing

---

[*]Work done while at ASAPP.

[1]https://github.com/asappresearch/scale-score

[2]For instance, OpenAI GPT-4 and Anthropic Claude support context sizes up to 32k and 100k tokens, respectively.

the explanation of model behaviors.

Current benchmark datasets for factual inconsistency detection predominantly feature short documents. In order to evaluate SCALE using a more realistic dataset with long documents, we introduce ScreenEval — a novel dataset designed to assess the factual inconsistency of summary sentences generated by humans, Longformer, and GPT-4 in comparison to actual long-form dialogues. ScreenEval encompasses 52 dialogues, averaging over 6,000 tokens per dialogue. The use of dialogue in this dataset poses a considerable unique challenges such as long-distance coreference resolution and significant noise between utterances. To the best of our knowledge, ScreenEval is the longest dialogue based dataset for factual inconsistency detection presently available.

In our experiments, we first show that SCALE outperforms and is better calibrated than baseline methods across various NLG tasks on the standard factual inconsistency detection benchmark TRUE (Honovich et al., 2022). We then assess accuracy, speed, and model explanation (via relevant text retrieval evaluation) on the new ScreenEval dataset for long document factual inconsistency detection. Our findings indicate that SCALE surpasses strong competitors in the majority of tests. The key contributions of this paper are:

- We introduce SCALE, a reference-free, NLI based factual inconsistency detection method with a novel chunking strategy for versatility across domains and extended documents.

- We show SCALE's broad applicability in NLG domains by attaining state-of-the-art performance on the TRUE benchmark.

- We build ScreenEval, a novel dataset designed for factual inconsistency detection in long dialogues, and then demonstrate SCALE's superiority in accuracy, efficiency, and model explanation evaluations on the dataset.

## 2 Related Work

**Factual Inconsistency Detection** There are two main directions in factual inconsistency detection: Natural language inference (NLI) based and question answering (QA) based methods. In NLI based methods, pretrained NLI models can be utilized to determine whether a given "premise" factually entails a "hypothesis." Although initial attempts

encountered challenges (Khot et al., 2018), recent advancements have shown that NLI models can effectively assess the factual consistency of generated text (hypothesis) with respect to a source (premise) (Utama et al., 2022). This progress can largely be attributed to addressing the granularity problem, which arises from the abundance of current NLI datasets predominantly comprised of short, single-sentence premises and hypotheses (Williams et al., 2017; Nie et al., 2019; Thorne et al., 2018a; Schuster et al., 2021a).

SCALE is an NLI based method and our findings indicate that utilizing larger premise chunks enhances efficiency and outperforms sentence decomposition. Although SeNtLI (Schuster et al., 2022) extended NLI based methods to longer documents, it adhered to the sentence decomposition assumption and focused solely on summarization tasks. SummaC (Laban et al., 2022) investigated various aggregation techniques for NLI scores obtained from sentence decomposition to generate overall summary scores. Meanwhile, SWING (Huang et al., 2023) developed a loss function to train models for improved NLI performance, yielding mixed outcomes.

In QA based methods, a question is first generated based on a summary sentence, and a QA system is used to give an answer. A summary is considered factually consistent if the generated answer significantly overlaps with the original summary (Durmus et al., 2020). Prior research focused on using different question generation strategies (Scialom et al., 2019) or overlap measures (Deutsch and Roth, 2021). In the experiments, we consider the most competitive QuestEval (Scialom et al., 2021a) and QAFactEval (Fabbri et al., 2021b).

**Detecting Factual Inconsistencies in Long Documents** Prior work on factual inconsistency performance in long source documents has been limited in scope. For example, ContractNLI (Koreeda and Manning, 2021) concentrates on legal documents, which differ significantly from dialogues in terms of challenges. Likewise, LongEval (Krishna et al., 2023) emphasizes human evaluation strategies for scoring, without considering dialogues. To our knowledge, this paper presents the first dataset for evaluating factual inconsistency in long-form dialogues, ScreenEval, thus addressing a substantial gap in the literature.

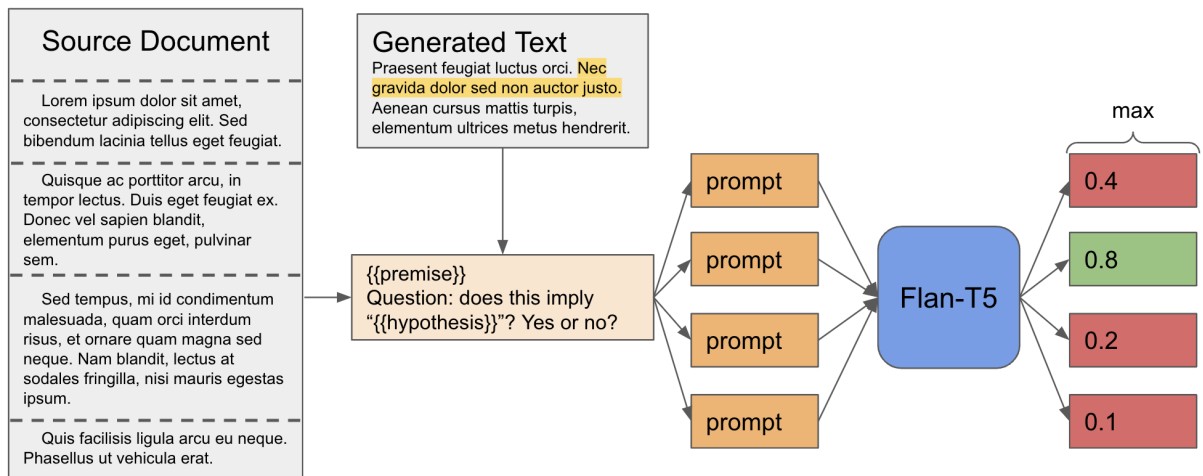

Figure 1: Chunking mechanism for SCALE to produce a score given a source document and generated text. The source document is broken into chunks (represented by dashed lines) and each chunk is fed into to a prompt as the premise. The highlighted generated text is fed into all prompts as the hypothesis. Each prompt is then run through Flan-T5 and the resulting logits are used to compute the entailment score.

## 3 SCALE

In this section we elaborate on our approach taken for our inconsistency detection model SCALE. Firstly, we formally define the use of chunks and NLI in SCALE, aiming at improving the accuracy and efficiency of the system. Secondly, we propose to explain the model output by retrieving the relevant source sentence for a target, and show how the relevant sentence retrieval can be improved through the use of chunks.

### 3.1 Chunking Mechanism for NLI based Model

Our approach uses NLI (Dagan et al., 2006) as a building block for factual inconsistency detection. An NLI model $M$ provides the relationship between a premise $p$ and a hypothesis $h$, $M(p, h)$ with probabilities of three labels: entailed, neutral, and contradictory. For example, given an NLI model $M(p, h)$, source document $D$ with a set of facts $F_D$, and a generated text $G$ with a set of facts $F_G$, if $F_G \subseteq F_D$ we would expect $M(D, G)$ to produce high entailment probability.

We define factual consistency between a generated text and source document as $F_G \subseteq F_D$. Canonical NLI models cannot be properly used for factual inconsistency detection because both $p$ and $h$ are commonly single sentences in NLI models, however in the factual inconsistency task their equivalents $D$ and $G$ almost always contain multiple sentences which $M$ cannot effectively handle, leading to an issue known as the granularity prob-

lem (Utama et al., 2022). To bypass the granularity problem, a natural generalization is to split both $D$ and $G$ into sentences and run $M$ pairwise on each of those sentences then using an aggregation function $f$ to generate the final entailment probability. Numerous papers have used this approach to generate competitive results (Schuster et al., 2022; Laban et al., 2022) however this generalization is hindered by a few shortcomings.

First, the sentence decomposition of $D$ and $G$ does not properly capture the context provided in $D$. By decomposing $D$ and $G$ into single sentences $D = (d_1, d_2, \ldots, d_i, \ldots, d_{|D|})$ and $G = (g_1, g_2, \ldots, g_j, \ldots, g_{|G|})$ and put into the model to evaluate as $M(d_i, g_j)$, the context and long term dependencies present in $D$ that may have factually supported $g_j$ very likely could not be represented in $d_i$. Multiple sentences (e.g., $\cup_{i \in \{1,3,6\}} d_i$) together in unison may be needed to support a single claim $g_j$. However, evaluating $g_j$ against each sentence individually $M(d_1, g_j)$, $M(d_3, g_j)$, $M(d_6, g_j)$ would likely lead to artificially low scores. Second, evaluating pairwise sentences of $D$ and $G$ is slow. It requires $|D| \cdot |G|$ model runs to obtain a final score for one sentence $g_j$.

SCALE poses a different solution to the granularity problem by decomposing $D$ into much larger chunks, which can be visualized in Figure 1. Formally, SCALE decomposes document $D$ into a set of $N$ chunks $C = c_1, c_2, \ldots, c_N$ such that $\cup_{c \in C} = D$. SCALE can handle chunks of arbitrary length only limited by memory requirements, thus drastically increasing the context win-

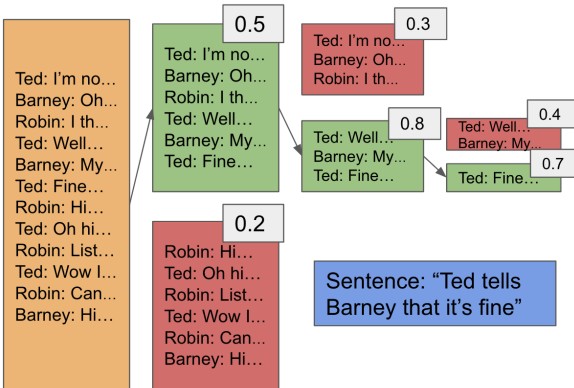

Figure 2: Visualization of SCALE's in context retrieval using chunks to find the most relevant source utterance given a sentence. Each chunk is scored by SCALE, as shown by the gray boxes.

dow provided to the model through the premise. The generated text $G$ is broken into sentences $G = (g_1, g_2, \ldots, g_j, \ldots, g_{|G|})$. We propose that decomposing $D$ using chunks rather than sentences does not negatively affect the granularity problem but rather enables superior contextual capture in model $M$, boosts accuracy, and requires significantly less model runs.

SCALE uses Flan-T5 (Chung et al., 2022) as a backbone NLI model $M$. SCALE obtains the probability that a chunk $c_i$ entails a generated sentence $g_j$ through the following steps. First, logits are obtained by prompting $M$ with the following: logits = $M$("{$c_i$} Question: does this imply '{$g_j$}'? Yes or no?"). The entailment probability between $g_j$ and $c_i$ is then calculated by

$$P_{\text{entail}} = \text{SoftMax}(\text{logits}["Yes"], \text{logits}["No"])[0].$$

To obtain the overall entailment score for a generated sentence $g_j$, the results are aggregated over all possible $c_i$ by,

$$\text{SCALE}(C, g_j) = \max_{i=1\ldots N}(P_{\text{entail}}(c_i, g_j))$$

to obtain a final measure of factual consistency.

### 3.2 Model Explanation via Relevant Source Text Retrieval

To produce explainable and interpretable scores for factual consistency and empower necessary human inspection, it is important to justify the score by retrieving relevant text from the source. Formally, the retrieval task involves finding the most relevant sentence $d_i$ in document $D$ with respect

to a hypothesis $h$. Using a new search tree approach enabled by chunking, SCALE is able to retrieve $d_i$ in context while using far fewer model runs than previous approaches. We use a greedy search tree approach that evaluates a hypothesis using SCALE against progressively smaller chunks to find the highly relevant text from the source document. For the following example, assume we use a binary search tree (BST) at each level, dividing the text into two chunks. This process can be visualized in Figure 2. Given a hypothesis $h$, we want to find the most relevant utterance $d_i$ in the source text. We begin by dividing the entire source document into two large chunks. SCALE is used to calculate the score between both chunks and the hypothesis $h$ and then use the higher scoring chunk as the new source text. The new source text is then divided into two chunks, and continues to descend in this manner until the chunk size becomes a single sentence or utterance $d_i$. The best scoring chunk is then chosen to be the supporting proof of the hypothesis from the source document.

This retrieval approach is able to significantly reduce the number of model calls needed to find the relevant text from a source document. Previous approaches commonly break the source document down by sentence which requires $O(n)$ model calls for a source document with $n$ sentences. Whereas our BST approach only needs $O(\log(n))$ model calls in order to find the most relevant utterance in the same source document of $n$ sentences.

Notice that we proposed the binary search scheme due to its simplicity and its connection to the popular binary search tree. In practice, dividing the source text into only two chunks might cause out of GPU memory issues. In this case, we could generalize the proposed approach into different chunk splits. For example, we could divide the remaining tokens into three chunks or larger for the search of each block until the model fits the chunk. We could also use different chunk sizes for different layers so long as it fits in the memory.

## 4 ScreenEval Dataset

We introduce a novel dataset for evaluating inconsistency detection on long form dialogues called ScreenEval. This dataset uses TV scripts and summaries pulled from the SummScreen (Chen et al., 2021) dataset. In addition to the provided human summaries, we generate summaries using Longformer and GPT-4 on 52 scripts from the Summ-

| Metric | Count |
|---|---|
| # of documents | 52 |
| # of summary sentences | 624 |
| avg. # of utterances per doc | 309 |
| avg. # of sentences per summary | 4 |
| avg. # of tokens per doc | 6073 |
| avg. # of tokens per summary | 101 |
| avg. # of tokens per summary sentence | 26 |
| # factually consistent sentences | 168 |
| # factually inconsistent sentences | 58 |
| avg. # of relevant utterances | 5 |

Table 1: Statistics for the ScreenEval dataset. These statistics use the Flan-T5 tokenizer

**Screen test set.** We then hire human annotators to classify the factual inconsistency of each summary sentence and identify relevant supporting utterances for factually consistent summary sentences. ScreenEval is released publicly. Details of how we use Longformer and GPT-4 and collect human annotation can be found in the Appendix A.

The SummScreen dataset is comprised of 2 sub datasets pulled from different sources Forever-Dreaming and TVMegaSite. We use the Forever-Dreaming subset of SummScreen, (SummScreen-FD) to create ScreenEval due to its manageable summary size and diversity of shows and genres, spanning a total of 21 genres. SummScreen-FD uses human-written gold summaries from Wikipedia and TVMaze. Table 1 shows statistics related to ScreenEval. Notably, the average number of tokens in a source document is 6,073, which, to the best of our knowledge, makes ScreenEval the longest dialogue based inconsistency detection dataset created. We provide 52 documents with an associated 624 summary sentences, 455 of which are artificially generated using Longformer and GPT-4. Summaries are kept at a high level, covering major plot points and character developments. This leads to shorter and more concise summaries that on average only run 101 tokens.

## 5 Experiments

### 5.1 Datasets

**TRUE** The TRUE benchmark contains 11 datasets from 4 different NLG tasks. We compare our approach to others on this dataset to show how SCALE performs across a multitude of NLG tasks on inconsistency detection. Notably, the average number of tokens per example in the TRUE benchmark is small, generally less than 512. Each dataset

in the TRUE benchmark is condensed down into a source document, generated text, and factual inconsistency label. The datasets are distributed across different tasks as shown in Table 2.

| Task | Examples | Datasets |
|---|---|---|
| Summarization | 5,245 | 5 |
| Dialogue | 10,613 | 3 |
| Fact Verification | 81,263 | 2 |
| Paraphrasing | 8,000 | 1 |

Table 2: Number of examples and datasets for each task in the TRUE benchmark.

Summarization datasets are from FRANK (Pagnoni et al., 2021), SummEval (Fabbri et al., 2021a), MNBM (Maynez et al., 2020), QAGS-CNNDM (Wang et al., 2020), and QAGS-XSum (Wang et al., 2020). Dialogue datasets include BE-GIN (Dziri et al., 2022), $Q^2$ (Honovich et al., 2021), and DialFact (Gupta et al., 2021). Fact Verification datasets are FEVER (Thorne et al., 2018b) and VitaminC (Schuster et al., 2021b). The Paraphrasing dataset is PAWS (Zhang et al., 2019b).

**ScreenEval** Our datasest ScreenEval compares the inconsistency detection ability of methods on long form dialogue. Details can be found in Sec. 4

### 5.2 Competitive Systems

TRUE provides 9 inconsistency detection baselines from 4 different inconsistency detection styles, namely n-gram based methods (token level F1), model based methods (BERTScore (Zhang et al., 2019a), BLEURT (Sellam et al., 2020), FactCC (Kryscinski et al., 2020), BARTScore (Yuan et al., 2021), CTC (Deng et al., 2021)), NLI based methods (ANLI (Honovich et al., 2022), SummaC (Laban et al., 2022)), and question answering (QA) based methods ($Q^2$ (Honovich et al., 2021), QuestEval (Scialom et al., 2021b)).

For ScreenEval we compare SCALE to 8 models which use NLI, QA, modern GPT systems, and older n-gram and semantic similarity methods. The baseline models consist of two NLI based sentence decomposition approaches seNtLI (Schuster et al., 2022), and SummaC$_{conv}$ (Laban et al., 2022), a state-of-the-art QA based model QAFactEval (Fabbri et al., 2022), a multidimensional QA model UniEval (Zhong et al., 2022), a semantic similarity model based method BERTScore (Zhang et al., 2019a), an n-gram overlap method ROUGE (Lin, 2004), and the two recent OpenAI models Chat-GPT, and GPT-4.

We also compare the performance of SCALE's search tree based relevant utterance retrieval with other recent retrieval models. We compare SCALE to the retrieval performance of Super-Pal (Ernst et al., 2020) which was shown to have superior retrieval capabilities in LongEval. We also compare against seNtLI (Schuster et al., 2022) which was designed to perform retrieval to identify factual inconsistencies over long documents.

For SCALE , we include three variants of Flan-T5 as the backbone, namely base, XL, and XXL.

## 5.3 Metrics

**Accuracy Evaluation** We compare the performance of methods primarily using four metrics, ROC_AUC score, Pearson correlation, Kendall_Tau correlation, and F1_Macro score. We employ the ROC_AUC score to quantify the ability of different methods in accurately identifying true positives and true negatives. Pearson and Kendall_Tau correlations show the relationship between methods and labels by measuring the correlations between the two. Finally the F1_Macro score is used to compare the continuous outputs of SCALE to the discrete outputs of GPT-4 and ChatGPT. To obtain an F1 score for SCALE we use the optimal threshold to convert its continuous output into discrete values.

**Efficiency Evaluation** We measure wall clock time in seconds for all of our experiments on ScreenEval. Wall clock time demonstrates how SCALE can be used efficiently in an online setting especially when compared to other models.

**Model Explanation Evaluation** We evaluate Calibration and Relevant Source Text Retrieval for model explanation.

**Calibration** is the measure of how close the pseudo-probability outputs of a model are to the actual probability of a correct prediction. For example, a well calibrated model that produces a score of $0.2$ would have a $20\%$ probability of being classified as 1. In this paper, we use Expected Calibration Error (ECE) (Guo et al., 2017) to compare the calibration of SCALE to other commonly used models.

Given model outputs spanning from 0 to 1, ECE separates the outputs into $K$ equally sized bins $B_k$ between 0 and 1 and takes the difference between accuracy $acc$ and confidence $conf$ in each one. The accuracy of a bin $B_k$ is the average amount of predicted labels that match true class labels in a bin, formally defined as

$$acc(B_k) = \frac{1}{|B_k|} \sum_{i \in B_k} \mathbf{1}(\hat{y}_i = y_i), \quad (1)$$

where $\hat{y}_i$ and $y_i$ are the predicted and true class labels for sample $i$. Confidence in a bin $B_k$ shows the average predicted score in a bin, formally defined as

$$conf(B_k) = \frac{1}{|B_k|} \sum_{i \in B_k} \hat{p}_i, \quad (2)$$

where $\hat{p}_i$ is the model output score for sample $i$.

Then the following equation is used to calculate ECE,

$$ECE = \sum_{k=1}^{K} \frac{|B_k|}{n} |acc(B_k) - conf(B_k)|, \quad (3)$$

using equation (1) and (2). A lower ECE indicates a better calibration.

**Relevant Source Text Retrieval** tests if each model could return the correct utterance identified as relevant by human labelers. We report the recall of retrieval results.

## 6 Results

### 6.1 TRUE

We first evaluate SCALE on the TRUE benchmark to confirm SCALE is NLG task agnostic and generalizes well to the factual inconsistency detection.

**Accuracy Evaluation Results** For the TRUE benchmark as shown in Table 3, SCALE$_{XXL}$ provides superior performance in 10 out of the 11 datasets, and SCALE$_{XL}$ achieves superior performance in 8 datasets compared to other non-SCALE models. Notably, other models were not previously able to perform well across all tasks, with $Q^2_{metric}$ having superior performance across 3 datasets and ANLI having superior performance across 5. These results demonstrate SCALE's ability to perform well across domains and against a large variety of model types.

**Model Explanation Evaluation Results** Not only does SCALE provide superior performance on the TRUE benchmark, but it is also highly calibrated across NLG tasks. Table 4 shows the ECE of multiple methods across the TRUE benchmark datasets. Note that a lower ECE is better. SCALE$_{large}$ provides the best calibration on

| Metric | FRANK | SummEval | MNBM | QAGS-C | QAGS-X | BEGIN | $Q^2_{ds}$ | DialFact | PAWS | FEVER | VitC | Avg$_{w/o\ VitC,FEVER}$ |
|---|---|---|---|---|---|---|---|---|---|---|---|---|
| $Q^2_{metric}$ | 87.8 | 78.8 | 68.7 | 83.5 | 70.9 | 79.7 | 80.9 | 86.1 | 89.7 | 88.4 | 81.4 | 80.7 |
| ANLI | 89.4 | 80.5 | **77.9** | 82.1 | 83.8 | 82.6 | 72.7 | 77.7 | 86.4 | 93.2 | 88.3 | 81.5 |
| SC$_{ZS}$ | 89.1 | 81.7 | 71.3 | 80.9 | 78.1 | 82.0 | 77.4 | 84.1 | 88.2 | ~~93.2~~ | ~~97.9~~ | 81.4 |
| F1 | 76.1 | 61.4 | 46.2 | 63.8 | 51.1 | 86.4 | 65.9 | 72.3 | 51.1 | 51.8 | 61.4 | 63.8 |
| BLEURT | 82.8 | 66.7 | 64.5 | 71.6 | 57.2 | 86.4 | 72.4 | 73.1 | 68.3 | 59.5 | 61.8 | 71.4 |
| QuestEval | 84.0 | 70.1 | 65.3 | 64.2 | 56.3 | 84.1 | 72.2 | 77.3 | 69.2 | 72.6 | 66.5 | 71.4 |
| FactCC | 76.4 | 75.9 | 59.4 | 76.4 | 64.9 | 64.4 | 63.7 | 55.3 | 64.0 | 61.9 | 56.3 | 66.7 |
| BART$_{score}$ | 86.1 | 73.5 | 60.9 | 80.9 | 53.8 | 86.3 | 64.9 | 65.6 | 77.5 | 64.1 | 63.2 | 72.2 |
| BERT$_{score}$ | 84.3 | 77.2 | 62.8 | 69.1 | 49.5 | 87.9 | 70.0 | 64.2 | 77.5 | 63.3 | 62.5 | 71.4 |
| SCALE$_{large}$ | 88.0 | 79.4 | 74.2 | 81.0 | 81.4 | 79.3 | 84.3 | 91.6 | 96.9 | 93.9 | 89.3 | 84.0 |
| SCALE$_{xl}$ | 89.8 | 83.3 | 73.2 | 82.1 | 84.3 | 77.8 | 86.3 | 91.0 | 91.9 | 94.5 | 91.5 | 84.4 |
| SCALE$_{xxl}$ | **90.8** | **91.5** | 73.1 | **85.2** | **85.3** | 79.2 | **86.6** | **93.4** | **96.7** | 94.8 | 92.7 | **88.1** |

Table 3: TRUE benchmark results. ROC_AUC scores multiplied by 100 for readability. Since $SC_{ZS}$ uses FEVER and VitC in training, these two datasets are excluded when computing the average.

| Metric | FRANK | SummEval | MNBM | QAGS-C | QAGS-X | BEGIN | $Q^2_{ds}$ | DialFact | PAWS | FEVER | VitC | Avg |
|---|---|---|---|---|---|---|---|---|---|---|---|---|
| UniEval | 0.333 | 0.238 | 0.173 | 0.26 | 0.315 | 0.507 | 0.153 | 0.396 | 0.335 | 0.084 | 0.242 | 0.276 |
| BERTScore | 0.2343 | 0.0730 | 0.1568 | 0.3200 | 0.4783 | 0.3119 | 0.1760 | 0.1388 | 0.1923 | 0.4217 | 0.0648 | 0.233 |
| SummaC$_c$ | 0.199 | 0.322 | 0.128 | 0.067 | **0.043** | 0.085 | 0.412 | 0.116 | 0.355 | 0.109 | 0.185 | 0.184 |
| SummaC$_z$ | 0.266 | 0.082 | 0.058 | 0.267 | 0.196 | 0.244 | **0.084** | 0.08 | 0.122 | **0.04** | 0.074 | 0.137 |
| SCALE$_{xl}$ | **0.093** | 0.243 | **0.026** | **0.060** | 0.116 | **0.069** | 0.286 | **0.071** | 0.060 | 0.209 | 0.150 | 0.126 |
| SCALE$_{large}$ | 0.179 | **0.043** | 0.215 | 0.144 | 0.117 | 0.149 | 0.103 | 0.117 | **0.019** | 0.058 | **0.049** | **0.108** |

Table 4: TRUE Calibration Results. ECE of each method on TRUE benchmark datasets (lower is better)

average and SCALE$_{XL}$ outperforms other non-SCALE models in calibration on over half of the datasets in the TRUE benchmark.

A visual example of the calibration results can be analyzed with the calibration curves in Figure 3. While most models are severely uncalibrated and underestimate the fraction of positives in Figure 3, SCALE is capable of sticking extremely close to the perfectly calibrated line. The closest model SummaC$_{Conv}$ can be seen overestimating positive examples before scores reach 0.4. We hypothesize that the large context window is the key to better calibration in SCALE as it includes more information. This makes the underlying NLI model less likely to be biased toward a specific range of tokens which leads to extreme confidence based on certain short text. To empirically justify this, we perform further experiments on the proposed ScreenEval dataset shown in Figure 4. We can observe that for chunk size < 400, the calibration score (the lower the better) is much higher than larger chunk size 500 to 1000. This shows that a larger chunk size could enable the NLI model to extract more useful information to provide appropriate confidence when making the prediction. We also use this knowledge to support our decision to use 512 tokens as our chunk size for all experiments in this paper. The enhanced calibration achieved by SCALE allows it to be more interpretable as a probability, making it a valuable tool for comparison tasks.

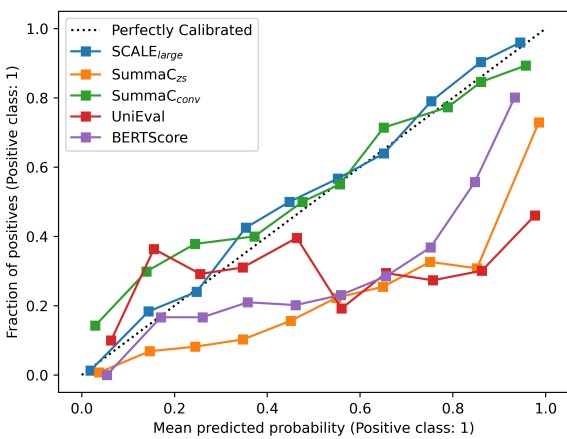

Figure 3: Calibration curves on the PAWS benchmark

## 6.2 ScreenEval

We then evaluate the performance of SCALE's chunking capabilities against other models in a long form dialogue setting using ScreenEval. We evaluate the factual inconsistency detection and relevant utterance retrieval of SCALE compared with other models and explore the unique problems posed by long form dialogue evaluation.

**Accuracy and Efficiency Evaluation Results**
We compare the factual inconsistency detection performance of multiple models on ScreenEval in Table 5. SCALE significantly outperforms other methods across all measures. While the state-of-the-art QA model QAFactEval was able to perform well on ScreenEval, SCALE$_{large}$ still showed superior performance across all metrics. Notably, even

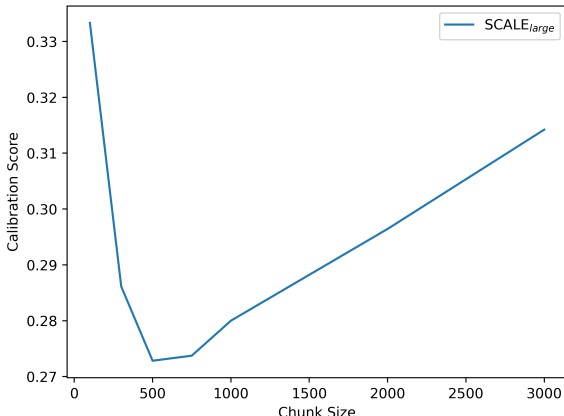

Figure 4: Effect of different chunk sizes on calibration performance on ScreenEval dataset.

| Metric | Pearson | Kendall-Tau | ROC-AUC | Time (s) |
|---|---|---|---|---|
| Rouge-1 | 0.120 | 0.093 | 57.6 | - |
| BERTScore | 0.130 | 0.076 | 56.2 | - |
| SummaC$_c$ | 0.030 | 0.011 | 50.9 | 1153 |
| UniEval | 0.066 | 0.094 | 57.6 | 1139 |
| seNtLI | 0.005 | 0.096 | 42.2 | 12688 |
| QAFactEval | 0.331 | 0.293 | 73.5 | 12132 |
| SCALE$_{base}$ | 0.28 | 0.24 | 69.5 | **678** |
| SCALE$_{large}$ | **0.391** | **0.322** | **76.1** | 1991 |

Table 5: Factual inconsistency detection results on ScreenEval. ROC_AUC is multiplied by 100 for readability.

SummaC$_{conv}$ and seNtLI, which are designed to deal with long documents, have poor performance on ScreenEval.

Along with its superior performance, SCALE is able to run faster than other LLM based methods on ScreenEval also shown in Table 5. For a fair comparison, we set the batch size to 1 for all models and run with all other default settings. We do not include BERTScore due to it's truncation of the document, making timing not comparable. Most notably QAFactEval, which was closest in performance to SCALE$_{large}$, was 6 times slower than SCALE$_{large}$ in wall clock time. Even faster though was SCALE$_{base}$ which was 17 times faster than QAFactEval while only achieving slightly worse performance across all metrics on ScreenEval, and outperforming all non-SCALE methods other than QAFactEval. The SCALE$_{base}$ model running at 1.1 seconds per score for long documents could realistically be used in an online setting to more accurately evaluate factual inconsistency.

Chunk size proves to have a large effect on the ability of SCALE's performance and time as seen in Figure 5. SCALE$_{large}$ sees a sharp increase in performance up until the chunk size is 1000 tokens

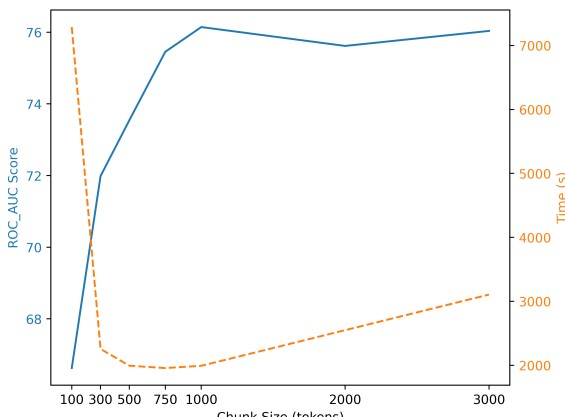

Figure 5: Effect of different chunk sizes on SCALE$_{large}$ performance and time on ScreenEval dataset. ROC_AUC score is multiplied by 100 for readability.

long. Similarly, there is a sharp decrease in model run time up until 1000 tokens. Figure 5 substantiates our approach to the granularity problem by illustrating that a larger number of tokens in the premise leads to a more effective method.

| Model | F1-score macro | Cost | Time (s) |
|---|---|---|---|
| SCALE$_{XL}$ | 73.86 | - | 5425 |
| SCALE$_{large}$ | 68.74 | - | **1991** |
| GPT-4 | **77.95** | $102 | 7255 |
| ChatGPT | 56.5 | $5 | 2544 |

Table 6: Compare SCALE with ChatGPT and GPT-4 on ScreenEval.

We additionally compare SCALE with ChatGPT and GPT-4 on ScreenEval in Table 6. Due to the discrete nature of GPT-4 and ChatGPT's output, we choose an ideal threshold for SCALE and compare macro F1-Scores on the ScreenEval dataset. While GPT-4 is able to outperform SCALE$_{XL}$ in macro F1-Score, SCALE shows to be significantly better in terms of time and cost. ChatGPT is more comparable in terms of time and cost to SCALE; however, there is a significant performance drop in macro F1-Score. ChatGPT is also limited by its 4096 token length limit at the time of writing and must use truncated conversations from ScreenEval. These results help us conclude that while GPT-4 has superior performance, SCALE is able to provide a faster, more affordable model that can be used locally in an online setting to produce continuous scores.

| Metric | Recall | Time (s) |
|--------|--------|----------|
| seNtLI$_{xl}$ | 0.343 | 35.35 |
| SuperPal | 0.406 | 6.13 |
| SCALE$_{large}$ | 0.463 | **4.67** |
| SCALE$_{xl}$ | **0.471** | 12.46 |

Table 7: Relevant source text retrieval results on ScreenEval. Time is measured in average number of seconds taken to retrieve the most relevant utterance on ScreenEval conversations.

**Model Explanation Evaluation Results** We now compare SCALE's BST retrieval approach with SuperPal and seNtLI. SCALE outperforms both in terms of time and performance as shown in Table 7. SCALE$_{XL}$ identifies the most relevant utterances correctly 47% of the time compared to 34% for seNtLI and 41% for SuperPal. SCALE's BST retrieval requires significantly fewer model calls, allowing it to pinpoint relevant utterances without having to score each one individually like the other methods. This results in higher retrieval recall for both SCALE$_{large}$ and SCALE$_{XL}$. Moreover, because SCALE$_{large}$ requires far less model calls it is able to provide faster results than Super-Pal or seNtLI without comprimising effectiveness. This enhanced performance shows how SCALE could be used in an online setting for fast and accurate results.

## 7 Conclusion

In this paper, we introduce a cutting-edge NLI based factual inconsistency detection method called SCALE. We show that SCALE is an NLG task agnostic factual inconsistency detection method by achieving state-of-the-art results across four distinct NLG tasks and 11 datasets within the TRUE benchmark. We show that across NLG tasks SCALE is also superior in calibration, providing scores that are interpretable and enables accurate comparison of scores. We introduce a new dataset called ScreenEval that is the first of its kind for long form dialogue factual inconsistency evaluation. SCALE is shown to significantly outperform all models other than GPT-4 in factual inconsistency detection on this dataset. However, we show that SCALE is significantly more cost effective and faster than GPT-4 for evaluation on ScreenEval. Moreover, we introduce a new retrieval strategy enabled by SCALE that significantly decreases the time to retrieve relevant utterances from a long document with increased recall accuracy.

## 8 Limitations

While SCALE performs well at inconsistency detection there are some limitations to this approach.

SCALE only uses the "Yes" and "No" logits to compute it's entailment score, however only using those two logits specifically could lead to loss of accuracy due to other information possibly flowing to similar tokens such as "yes", "no". Using logits for scoring purposes may cause a loss of information to other similar logits.

Finally, even though SCALE is able to achieve better calibration in the aggregate, it still struggles with calibration on certain tasks and this can even vary by model size. Consistent calibration of scoring methods across NLG tasks should be a goal for future methods.

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

## Appendix

## A Details of Construction of ScreenEval Dataset

We create summaries for ScreenEval using two summarization models, GPT-4 and Longformer, as well as human generated summaries. The models that we use for summarization are designed to have a large token context window, giving them both the ability to globally attend over the long source dialogues we provide. Below, we will first explain how to leverage Longformer, GPT-4 to generate summaries, and then explain the process and the cost of human labeling.

### A.1 Building the Dataset

We first generated longformer summaries for each script in the test set. To keep the annotation task reasonable, and to filter out any rambling summaries, we filter out any TV scripts with a longformer or human summary that had more than 6 sentences or only 1 sentence. We still preserve 52 of the TV scripts by doing this, as the median number of summary sentences in a longformer summary was 4 and a human summary was 3. In order to meet GPT-4's token limit requirements, from the remaining TV scripts we chose all that had less than 8,100 tokens. Our final dataset consists of 52 TV scripts that have an average length of 6073 tokens with 624 summary sentences.

### A.2 Longformer

We use the same baseline model as in Summ-Screen to generate summaries in ScreenEval, a Longformer model finetuned on SummScreen-FD's training set. This model uses a transformer based sequence to sequence architecture to globally attend over the entire dialogue. Longformer is able to take as many as 16384 tokens as input.

### A.3 GPT-4

GPT-4 is a large language model that has shown near human level performance in a wide variety of tasks, including summarization. We task GPT-4 to summarize each document in ScreenEval using the prompt "Summarize in 5 sentences or less: ". To accommodate for the roughly 8k token limit on the GPT-4 model, we specifically select documents in ScreenEval that are under 8k tokens.

### A.4 Human Annotation

We lable ScreenEval using workers from amazon mechanical turks with the promtp shown in Figure 6 and Figure 7. For each task, a worker is presented with a TV script from ScreenEval along with a highlighted summary sentence. The worker is instructed to first read through the source dialogue. Then, the worker is instructed to click either a Yes or No button to indicate whether the highlighted summary sentence is consistent with the source document. Each utterance in the TV script will be presented alongside a check box where if the worker chose "Yes" to the consistency question, they will be asked to select the relevant utterances that led to their answer.

We had 3 human annotators label each instance, and 61% of the time all three annotators agreed. Workers were paid 0.27 per task. We ensured the quality of annotators through a number of methods. First, we filtered annotators to just those located in the US and Canada to increase the chances of high fluency in English on our reading comprehension task. Additionally the workers had to have an MTurk "Master" qualification, greater than a 95% task approval rate, and greater than 5000 tasks approved. The dataset was labeled in batches of 30 at a time and closely monitored by the authors. Workers were only rejected if they did not list relevant utterances as instructed or listed non existent utterances, and these workers were able to dispute rejection via email.

## B Prompts Used for GPT

### B.1 ChatGPT/GPT-4

{Dialogue}
Question: does the previous conversation factually imply "{Summary Sentence}"?
Answer Yes or No.

## Summary

*(focus on highlighted span)*:

In the latest episode of National Geographic's Mars series, the crew of the Daedalus spacecraft face a series of challenges as they attempt to establish a base on the red planet. ==The crew's commander, Ben Sawyer, is injured during the landing and the team must race against time to reach their base camp and save his life.== Along the way, they face harsh conditions and equipment failures, highlighting the dangers and sacrifices involved in space exploration. The episode also features interviews with real-life astronauts and experts, discussing the physical and psychological challenges of long-duration space missions. Ultimately, the crew must come to terms with the loss of their commander and continue their mission to establish a human presence on Mars.

## Source Document

**\*\*Scroll\*\* within the text box to see the entire source document**

0: Amelie Durand: Previously on Mars.
1: Ben Sawyer: We knew Mars wouldn't welcome us with open arms.
2: Mae: Warning, systems offline.
3: Robert Foucault: Permission to switch from primary to backup.
4: Ben Sawyer: Do it.
5: Hana Seung: Prepare for retropropulsion.
6: Ben Sawyer: We were ready to give everything to get there.
7: Hana Seung: Mission control, this is Daedalus, we're looking at a red planet.
8: [Cheers]

Figure 6: Prompt provided to mturk labellers 0

5: Hana Seung: Prepare for retropropulsion.
6: Ben Sawyer: We were ready to give everything to get there.
7: Hana Seung: Mission control, this is Daedalus, we're looking at a red planet.
8: [Cheers]
9: Ben Sawyer: And we knew we'd have to fight even harder if we wanted to stay.
10: Marta Kamen: How far did we overshoot?
11: Sam: 75.3 kilometers.
12: Robert Foucault: Mission control confirmed the rover is 2,000 kilos over maximum payload.
13: Hana Seung: The odds are we won't make it.
14: Ben Sawyer: But no matter what happened, we'd do everything we could to make the workshop module our new basecamp.
15: Because the thing that kept us pushing, it wasn't just inside us. Our species has always been driven to build beyond what we know. We migrated across all of earth. Created settlements, constructed

## Questions

1. Is the summary sentence factually consistent with the source?

○ Yes   ○ No

2. ==**IF YOU ANSWERED YES**== list the utterance numbers that support your conclusion below, separated by commas. Try to list MORE THAN ONE.

Type NUMBERS here (i.e. 3, 89, 215)...

[ Submit ]

Figure 7: Prompt provided to mturk labellers 1