# OpenReview forum: "Fast and Accurate Factual Inconsistency Detection Over Long Documents"
_EMNLP/2023/Conference — EMNLP 2023 Main_

### Official Review · Reviewer_xTxN · 2023-08-05

**Soundness:** 4

**Excitement:**

2: Mediocre: This paper makes marginal contributions (vs non-contemporaneous work), so I would rather not see it in the conference.

**Paper Topic And Main Contributions:**

Propose a NLI based model to detect factual inconsistency in long documents, namely SCALE (Source Chunking Approach for Large-scale inconsistency Evaluation). Propose a valuable datasets to evaluate the model's performance on identifying factual inconsistency in long documents. Propose to leverage chunking method to retrieve relevant sentences as supporting evidence to support the model's decision.

**Questions For The Authors:**

1. how do you decide size of the chunk? Is it decided just by the context window of a model? One way to improve the technical innovation of this paper is to automatically decide the size of the chunking window since using huge chunking window can include irrelevant information.

2. on line 281-285, what if the model can not fit the chunk since here you claim to break the document into two chunks.



**Reasons To Accept:**

1. the motivation of detecting factual inconsistency in long documents is clear and promising.
2. the paper is well-written and easy to follow
3. proposed a new dataset named ScreenEval which is the longest dialogue based dataset for factual inconsistency detection.
4. achieves better performance comparing to sentence based nil.



**Reasons To Reject:**

1. The proposed method lacks innovation. The core idea of scale is to break a document into large chunks different from previous methods which break a document into sentences.

2. The "interpretation via Relevant Source Text Retrieval" is merely an adhoc straight-forward application of the NLI model and a binary search tree algorithm.

3. some details are missing to support the quality of the proposed dataset. Did you have multiple annotators to label each instance? If so, what is the agreement between annotators? If not, how did you select the annotators and how did you ensure the quality of their annotations. what is the payment for the annotators?

**Reproducibility:**

3: Could reproduce the results with some difficulty. The settings of parameters are underspecified or subjectively determined; the training/evaluation data are not widely available.

**Reviewer Confidence:**

3: Pretty sure, but there's a chance I missed something. Although I have a good feel for this area in general, I did not carefully check the paper's details, e.g., the math, experimental design, or novelty.

---

> ### Author Rebuttal · Authors · 2023-08-29
>
> - On Core Idea and Innovation.
> \
> The literature for factual consistency detection currently relies on a misguided assumption that sentence based decomposition is the superior approach to decomposing a document for testing factual consistency, when in fact it severely degrades performance of all models. SCALE identifies this flaw in the literature, proposes an accessible solution to the problem (chunking), and rigorously tests this new solution over both old (TRUE benchmark) and new (ScreenEval) datasets. Additionally, we are the first to test current factual consistency state of the art metrics on long form documents (>512 tokens) which is a vital new area given the ever growing context lengths of modern LLMs such as GPT-4. We show that all other state of the art methods severely underperform on these long documents while SCALE is able to maintain its performance by using chunking.
> \
> The simple and effective method of chunking significantly outperforms other complex factual consistency detection systems and is innovative. We believe that SCALE deserves to be included in the conference preceding so that many people can leverage the proposed method to build a practical system to solve this high-impact problem.
>
> - On Relevant Source Text Retrieval
> \
> We thank the reviewer for this observation, and note that it is possible to do NLI based relevant source text retrieval. However, previous approaches such as seNtLI, which uses retrieval as a central part of their paper, suffer from efficiency (each sentence must be individually scored to find the most relevant source) and underperformance due to out of context scoring (again, acting under the flawed assumption that the surrounding text doesn’t influence the factual consistency score). Chunking naturally enables the use of binary search trees which naturally solves both of these problems by using less model calls and selecting the relevant source in context of the entire story, keeping in theme with the paper. The results of this change can be seen in Table 7. We will make the motivation for using search trees as an algorithm more clear in the final version of the paper.
>
> - On what if the model can not fit the chunk
> \
> We proposed the binary search scheme (i.e., two chunks) due to simplicity and its connection to the popular binary search tree. But notice that we could generalize it into different chunk sizes. For example, we could divide the remaining tokens into three chunks or larger for the search of each block until the model fits the chunk. We could also use different chunk sizes for different layers so long as it fits in the memory. We will explain this concept in the final version of the paper.
>
> - On the details of the ScreenEval dataset
> \
> We had 3 human annotators label each instance, and 61% of the time all three annotators agreed. Workers were paid $0.27 per task. We ensured the quality of annotators through a number of methods. First, we filtered annotators to just those located in the US and Canada to increase the chances of high fluency in English on our reading comprehension task. Additionally the workers had to have an MTurk “Master” qualification, greater than a 95% task approval rate, and greater than 5000 tasks approved. The dataset was labeled in batches of 30 at a time and closely monitored by the authors. Workers were only rejected if they did not list relevant utterances as instructed or listed non existent utterances, and these workers were able to dispute rejection via email. Additional screenshots and details of the MTurk annotation task can be found in Appendix A.4. We will be adding all the details in the final version.
>
> - On the selection of chunk size
> \
> We analyzed the effect of chunk size in Section 6.2(L528 - L536). In sum, chunk sizes between 500 to 1000 gives the best overall performance and reasonably fast inference time. Thus we set the chunk size to be 512 throughout our experiments. As suggested by the reviewer, using a huge chunking window might include irrelevant information and our experiment as shown in Figure 4 justified that the performance will drop a bit if the chunk size is too large.

---

### Official Review · Reviewer_p7Jm · 2023-08-05

**Soundness:** 3

**Excitement:**

3: Ambivalent: It has merits (e.g., it reports state-of-the-art results, the idea is nice), but there are key weaknesses (e.g., it describes incremental work), and it can significantly benefit from another round of revision. However, I won't object to accepting it if my co-reviewers champion it.

**Paper Topic And Main Contributions:**

The paper proposes a new method called SCALE for detecting factual inconsistencies between generated text and source documents. SCALE is a task-agnostic model that uses Natural Language Inference (NLI) and a new chunking strategy to efficiently handle long texts. The paper also presents ScreenEval, a new dataset designed to test the factual inconsistency of generated summaries in comparison to actual long-form dialogues. SCALE is demonstrated to outperform existing methods and to perform effectively on various tasks and long inputs, with ScreenEval serving as a realistic testing ground for long document inconsistency detection.

**Reasons To Accept:**

1.	The proposed SCALE based on chunking is simple and effective in efficiently handling long texts.
2.	The authors demonstrate that SCALE outperforms existing methods on both standard benchmarks and the new ScreenEval dataset. This superior performance indicates that the proposed method provides an effective solution to the problem of factual inconsistency detection.
3.	The paper presents ScreenEval, a novel dataset for evaluating factual inconsistency detection, specifically designed for long dialogues. This dataset can serve as a valuable resource for future research in this area, enabling a more realistic and extensive evaluation of methods.


**Reasons To Reject:**

1.	Some implementation details can be better described, such as the techniques of decomposing documents into chunks.
2.	The paper mentions calibration scores in the introduction section but omits them in the method section. The reason why the proposed method can yield better calibration scores is not explicitly stated, which may lead to confusion or unanswered questions for the readers.
3.	The figures are a bit blurry and could be improved.


**Reproducibility:**

4: Could mostly reproduce the results, but there may be some variation because of sample variance or minor variations in their interpretation of the protocol or method.

**Reviewer Confidence:**

3: Pretty sure, but there's a chance I missed something. Although I have a good feel for this area in general, I did not carefully check the paper's details, e.g., the math, experimental design, or novelty.

---

> ### Author Rebuttal · Authors · 2023-08-29
>
> - On implementation details
> \
> The most basic form of chunking as described on L231-L239 is performed on a word by word basis such that the union of all the chunks is equivalent to the source document. This can also be visualized as the dashed lines in the Source Document in Figure 1. We will be revising the final version to make these more clear.
>
> - On the reason for good calibration performance
> \
> We thank the reviewer for pointing this out. Our main contribution is to show SCALE could further extend the applicability of NLI-based methods. In addition to accuracy, SCALE comes with various good properties and “better calibration” is one of them. We saved our discussion of Calibration for the Experiments section under L421-L448, but we will be re-organizing the paragraph to make it more clear in the final version.   \
> We hypothesize that the large context window is the key to better calibration in SCALE as it includes more information. This makes the underlying NLI model less likely to be biased toward a specific range of tokens which leads to extreme confidence based on certain short text. To empirically justify this, we perform further experiments on the proposed ScreenEval dataset. We change the chunk size from 100 to 1000 and observed that for chunk size < 400, the calibration score (the lower the better) is much higher (e.g., size=100 Score = .332, size = 300, Score = .305) than larger chunks (e.g., size = 500, Score = .272, size = 1000 Score = .278). This shows that a larger chunk size could enable the NLI model to extract more useful information to provide appropriate confidence when making the prediction.  \
> In sum, larger chunks not only help to improve accuracy (as shown in Fig. 4), it could also improve the calibration. We will be adding these new experiments and analysis in the final version.
> \
> - On the Figure Clarity
> \
> We thank the reviewer for pointing this out! We have high quality PDF formatted images that have already been swapped in for the next version of the paper.

---

### Official Review · Reviewer_2Gpy · 2023-08-05

**Soundness:** 3

**Excitement:**

4: Strong: This paper deepens the understanding of some phenomenon or lowers the barriers to an existing research direction.

**Paper Topic And Main Contributions:**

This work introduces a cutting-edge NLI based factual inconsistency detection method called SCALE.
SCALE is an NLG task agnostic factual inconsistency detection method by achieving state-of-the-art results across four distinct NLG tasks and 11 datasets within the TRUE benchmark.

**Reasons To Accept:**

- This work introduces SCALE, a reference-free, NLI based factual inconsistency detection method with a novel chunking strategy for versatility across domains and extended documents.
- SCALE is an NLG task agnostic factual inconsistency detection method by achieving state-of-the-art results across four distinct NLG tasks and 11 datasets within the TRUE benchmark.
- This work introduces a new dataset called ScreenEval for long form dialogue factual inconsistency evaluation.
- SCALE is shown to significantly outperform all models other than GPT-4 in factual inconsistency detection on this dataset. It is also significantly more cost effective and faster than GPT-4 for evaluation on ScreenEval.


**Reasons To Reject:**

- It would be better to publicly release code and dataset.

**Reproducibility:**

4: Could mostly reproduce the results, but there may be some variation because of sample variance or minor variations in their interpretation of the protocol or method.

**Reviewer Confidence:**

3: Pretty sure, but there's a chance I missed something. Although I have a good feel for this area in general, I did not carefully check the paper's details, e.g., the math, experimental design, or novelty.

---

> ### Author Rebuttal · Authors · 2023-08-29
>
> - On the release of code and dataset \
> We thank the reviewer for the kind remarks and note we will release the code and dataset publicly in the final version. This is also stated explicitly in L24 and L25 in the abstract.

---

### Meta-Review · Area_Chair_QbVh · 2023-09-19

**Recommendation:** 4

**Metareview:**

This work proposes a method for detecting factual inconsistencies in long documents by applying Natural Language Inference (NLI) to chunks of the document. Additionally, the authors introduce a new dataset, ScreenEval, which is a dialogue-based dataset designed for factual inconsistency detection.

The method is simple, scalable, yet outperforms the baseline approaches. Furthermore, the new dataset is expected to be a valuable resource for the research community. Overall, I don't see any significant issues with the work, aside from minor points related to clarity.

---

### Decision · Program_Chairs · 2023-10-07

**Decision:**

Accept-Main

**Comment:**

This work proposes a method for detecting factual inconsistencies in long documents by applying Natural Language Inference (NLI) to chunks of the document. Additionally, the authors introduce a new dataset, ScreenEval, which is a dialogue-based dataset designed for factual inconsistency detection.

The method is simple, scalable, yet outperforms the baseline approaches. Furthermore, the new dataset is expected to be a valuable resource for the research community. Overall, I don't see any significant issues with the work, aside from minor points related to clarity.